# The Role of Substrate on Thermal Evolution of Ag/TiO_2_ Nanogranular Thin Films

**DOI:** 10.3390/nano11092253

**Published:** 2021-08-31

**Authors:** Vincenzo Balzano, Emanuele Cavaliere, Mattia Fanetti, Sandra Gardonio, Luca Gavioli

**Affiliations:** 1Interdisciplinary Laboratories for Advanced Materials Physics (i-LAMP), Dipartimento di Matematica e Fisica, Università Cattolica del Sacro Cuore, Via Musei 41, 25121 Brescia, Italy; vincenzo.balzano@unicatt.it (V.B.); emanuele.cavaliere@unicatt.it (E.C.); 2Materials Research Laboratory, University of Nova Gorica, Vipavska 11c, 5270 Ajdovščina, Slovenia; mattia.fanetti@ung.si (M.F.); sandra.gardonio@ung.si (S.G.)

**Keywords:** nanogranular composite thin film, substrate driven properties, morphology, optical absorption, Ag/TiO_2_

## Abstract

In multicomponent thin films, properties and functionalities related to post-deposition annealing treatments, such as thermal stability, optical absorption and surface morphology are typically rationalized, neglecting the role of the substrate. Here, we show the role of the substrate in determining the temperature dependent behaviour of a paradigmatic two-component nanogranular thin film (Ag/TiO_2_) deposited by gas phase supersonic cluster beam deposition (SCBD) on silica and sapphire. Up to 600 °C, no TiO_2_ grain growth nor crystallization is observed, likely inhibited by the Zener pinning pressure exerted by the Ag nanoparticles on the TiO_2_ grain boundaries. Above 600 °C, grain coalescence, formation of However, the two substrates steer the evolution of the film morphology and optical properties in two different directions. anatase and rutile phases and drastic modification of the optical absorption are observed. On silica, Ag is still present as NPs distributed into the TiO_2_ matrix, while on sapphire, hundreds of nm wide Ag aggregates appear on the film surface. Moreover, the silica-deposited film shows a broad absorption band in the visible range while the sapphire-deposited film becomes almost transparent for wavelengths above 380 nm. We discuss this result in terms of substrate differences in thermal conductivity, thermal expansion coefficient and Ag diffusivity. The study of the substrate role during annealing is possible since SCBD allows the synthesis of the same film independently of the substrate, and suggests new perspectives on the thermodynamics and physical exchanges between thin films and their substrates during heat treatments.

## 1. Introduction

The functionalities of metal/oxide thin films depend on the physical properties of the constituents and on the microscopic interactions at the film/substrate interface. Several factors such as the morphology, composition, crystalline phase and porosity of the materials may be controlled by synthesis methods such as cold gas dynamic spray [1], jet printing [2], supersonic cluster beam deposition (SCBD) [3,4], sol-gel methods [5,6,7,8], magnetron sputtering [9,10,11,12] and dip-coating [13]. Post-deposition annealing treatments can further tweak the film behaviour, bringing the substrate properties into play. To properly identify the substrate influence on the film behaviour as a function of temperature, however, one should be able to deposit the same film independently on the support type. Such opportunity is provided by spray synthesis techniques such as SCBD, where the film building blocks are maintained during landing and the film morphology at the nanoscale is determined only by the NP assembly [14,15,16,17,18].

A paradigmatic nanogranular model system to investigate the temperature effects is constituted by a nanostructured film composed of metal and oxide nanoparticles (NPs). Metal/titanium dioxide (TiO_2_) nanostructures have in fact unique properties for applications in photocatalysis [19,20,21,22,23], antibacterial coatings [18,24,25], sensors [26,27,28,29], solar cells [30,31,32] and energy storage [33,34]. The addition of NPs of different materials to a host matrix of TiO_2_ usually results in an increase of the stable temperature range of the composite film, due to the pinning force introduced by the NPs that hampers the coalescence and crystallization [35,36]. For example, the addition of small silica and alumina particles in TiO_2_ allows for a stabilization of the anatase phase up to 800 °C [37]. In this perspective, SCBD, which allows the deposition of clusters composed of both the host material and the pinning particles [24], is a promising technique for the deposition of multielement heat-resistant thin films.

Temperature may also lead to structural modifications and variation of the optical response of the system. In Ag/TiO_2_ nanocomposites, Ag interdiffusion and the NPs coalescence inside the film results in the formation of nanometre-sized Ag clusters on the film surface [9,38,39,40,41]. Ag/TiO_2_ nanocomposites exhibit an optical absorption band around 400–500 nm, which is usually associated to the plasmon resonance of Ag NPs [39,42,43,44]. The wavelength and shape of such absorption feature can be drastically modified by heat treatments: for example, in Ag-TiO_2_ coatings prepared by atom beam sputtering, annealing at 600 °C causes a redshift and a width increase of the absorption band [41]. Modification of the Ag morphology and distribution within the film would hence affect also the multicolour photochromism [45,46], the surface-enhanced Raman scattering [47,48] and the antimicrobial activity of such systems [24]. Since the amount, dimension and distribution of Ag NPs determine the optical properties of the films [49,50,51], understanding the mechanics of Ag segregation and clustering is a necessary step towards the synthesis of new materials with specific absorption and transmission bands. This is especially important for applications in photocatalysis: for example, the inclusion of Ag NPs can be used to extend the photoresponse of TiO_2_ to the visible range of the spectrum [49,50,51].

In the study of the temperature dependent morphological changes of Ag/TiO_2_ nanostructured materials, however, the role of the substrate was only considered in a single case, where a large decrease in optical absorption in the 400–500 nm wavelength range after an annealing at 450 °C was attributed to the diffusion of Ag atoms into the substrate [52]. This indicates a lack of a deep understanding of the substrate role for this type of nanocomposites, that would be of paramount importance to fully understand the formation of tailor-sized Ag clusters in the development and engineering of applications. This open issue asks for a synthesis technique of nanogranular systems allowing the deposition of films with the same properties independently of the substrate characteristics.

SCBD presents several advantages in terms of deposition rate, control over the film thickness and roughness and mass selection of the particles by aerodynamic effects, which allow to fabricate nanostructured system with the desired structural properties in an effective way [14,53,54,55]. Furthermore, SCBD allows for the simultaneous deposition of both metal and oxide NPs, maintaining the nanostructured nature of the building blocks, the AgTiO_2_ NPs. Since the films deposited by SCBD are the same independently of the substrate on which they are deposited, this technique is suitable to investigate the role of the substrate on the evolution of the films during thermal annealing.

In this work, we exploit SCBD to deposit identical Ag/TiO_2_ nanogranular thin films on fused silica (SiO_2_) and single-crystal sapphire (Al_2_O_3_) at room temperature (RT), and we analyse the temperature-dependent evolution of their structural and optical properties. Silica and sapphire were chosen due to the almost constant (a 3% variation) refractive index in the 320–2000 nm range [56], chemical and thermal stability up to 900 °C, oxidized surface termination, comparable surface roughness and thermal conductivity (which differs by an order of magnitude). We show that the Ag distribution inside the films evolves in a completely different way depending on the type of substrate.

## 2. Materials and Methods

Supersonic cluster beam deposition is a bottom-up, physical approach to NP synthesis which relies on the plasma ablation of a target rod of the desired material to produce nanostructured films. The principles of the SCBD are described in detail elsewhere [14,15,24,57,58]. In our experiments, a 99.99% purity Ag/Ti rod with a nominal 50–50 Ag-Ti wt.% (HMW Hauner GmbH) was employed as the starting material. The corresponding nominal atomic concentration is 30–70% Ag-Ti. The Ag/TiO_2_ (Ag-Ti 50–50) nanogranular films were grown at RT on two different 10 × 10 mm^2^ substrates, a 0.48 mm thick slab of (0001) α-Al_2_O_3_ single crystal and a 1 mm thick of amorphous fused silica, respectively (both provided by MaTeck GmbH, Germany). The substrates were cleaned with acetone and isopropyl alcohol before any deposition. The films were deposited in medium vacuum (base pressure = 1 × 10^−6^ mbar) and then exposed to air in order to obtain complete titanium oxidation to TiO_2_ [24]. The nominal thickness and deposition rate of the films were evaluated using a quartz microbalance, while the actual film thickness, measured on an edge by atomic force microscopy (AFM), was 50 ± 2 nm. The total deposition time was 45 min. The annealing steps of 1 h were performed in an oven at standard atmosphere and pressure at increasingly high temperatures (200 °C to 900 °C). During the annealing process, the samples were placed on alumina slabs in order to minimize heating by contact with the oven surface. The heating of the films is then achieved by contact with the air and by blackbody (infrared) irradiation from the oven cavity, which minimizes the influence of the substrate thickness on the heating rate. The cooling process was provided inside the annealing oven with an estimated cooling rate of 0.21 °C/s.

The surface morphology of the films was investigated via AFM (Park NX10) in tapping mode. A PPP-XYNCSTR-50 cantilever with 10 nm nominal tip radius and a typical resonating frequency of 160 kHz was employed. The raw AFM images were analysed with Gwyddion, applying a watershed algorithm to deduce the grain size.

The film transmittance and reflectance were obtained in the UV-Vis-NIR range (200–2500 nm) with a Perkin-Elmer Lambda 950 spectrophotometer (Perkin Elmer Inc., Waltham, MA, USA) equipped with a 150 mm integrating sphere.

The Raman spectra were acquired on a Renishaw inVia microscope equipped with a 633 nm laser (Renishaw RL633, maximum nominal power of 10 mW). A 50× objective and a laser beam intensity of 1% were used.

The film composition was evaluated by X-ray photoelectron spectroscopy (XPS) at the end of the annealing cycle using a dual anode (Mg-Al) X-Ray source and a Phoibos 100 SPECS analyser. The samples were introduced directly after deposition in the XPS system without any further treatment. The incident photon energy was 1253.6 eV. The corresponding photoelectron sampling depth for the elements composing the film is up to 5 nm. A field emission scanning electron microscopy (SEM) apparatus (FESEM, JEOL JSM 7100F TTLS) operating at a base pressure of <1 × 10^−4^ Pa and equipped with secondary and backscattered electron detectors was used for imaging of the film. The spatial distribution and the relative concentration of Ag and Ti were evaluated by energy dispersive X-ray spectroscopy (EDX) (spectrometer: Oxford X-Max 80) installed on the same microscope. Prior to the SEM/EDX analysis, the samples were coated with a 6 nm thick carbon film to avoid charging. A schematic diagram of the deposition and annealing process of the samples, as well as the characterization steps employed to investigate their properties, is available in the electronic Appendix A.

## 3. Results and Discussion

### 3.1. Morphology of the Ag/TiO_2_ Films

The morphological properties of the Ag/TiO_2_ films were investigated at RT by AFM on the as-deposited films and after every annealing step on both silica- and sapphire-deposited films. Films deposited by SCBD are unaffected by the substrate at deposition, since their structure is only determined by the properties of the Ag/TiO_2_ NPs and their aggregation on the substrate. In particular, the landing energy per atom allows for the conservation of the NPs integrity [14,59,60,61]. It is worth recalling that the NPs forming the nanogranular film are constituted by Ag nanocrystals partially embedded into an amorphous TiO_2_ matrix, as reported in details elsewhere [4,24]. Figure 1 shows the evolution of the surface morphology of the two films obtained by AFM. The as-deposited films exhibit the usual granular and homogeneous morphology previously reported [4,16,24,25]. For annealing temperatures up to 500 °C, no significant change in the surface structure or grain size can be observed. Above 500 °C, the formation of new island-like structures on the surface can be observed on the sapphire-deposited sample. The size of these structures increases gradually with subsequent annealing steps. At 600 °C, most of the islands are irregularly-shaped, with a mean width of 128 ± 42 nm and a mean height of 82 ± 68 nm (as deduced from AFM data). At 900 °C, the structures are roughly cylindrical, with a mean width of 570 ± 12 nm (deduced from SEM, see Appendix A of the Appendix A) and a mean height of 360 ± 8 nm (as deduced from AFM data, see Appendix A of the Appendix A). The presence of such islands is compatible with the migration and clustering of Ag [38]. On the silica-deposited sample, however, no such structures can be observed at any stage of the annealing process. In addition to the migration and surfacing of Ag, the annealing of the films for temperatures higher than 600 °C also results in a noticeable coarsening of the Ag/TiO_2_ grains on both substrates. Moreover, on the sapphire deposited film one observes the formation of 50 ± 1 nm deep pits, thus reaching the substrate.

The quantification of the morphology evolution is obtained through the normalized grain size distribution (GSD) shown in Figure 2a for the silica-deposited film as a function of the annealing temperature. For each GSD (obtained from three different images for a total of 1200–1500 particles per distribution), the grain size was measured from the grain projected surface area, a value that is influenced by tip-sample convolution effects. For a comparison with the actual NPs size distribution, see Appendix A of the Appendix A. The GSD follows a lognormal shape typical of films deposited by SCBD [14]. The evolution of the mode and standard deviation of the GSD patterns are shown in Figure 2b, while the root mean square roughness (RMS) of the surface is given in Figure 2c. For temperatures up to 500 °C, the GSD maintains the lognormal shape with the most recurring grain radius being 20 ± 1 nm, indicating that the film is nearly unaffected by the annealing process. This behaviour is different from pure TiO_2_ NPs films, where grain coarsening is taking place for temperatures as low as 300 °C [62,63,64,65].

The thermal stability of the grains in the present Ag-TiO_2_ nanocomposite is compatible with the hypothesis of a Zener pinning pressure exerted by the Ag NPs on the grain boundaries of the TiO_2_ NPs [35]. This effect, related to the presence of a finely dispersed distribution of secondary particles or defects within a granular material, is known to prevent or delay the temperature-induced nucleation and recrystallization of the principal grains in a variety of polycrystalline materials [66]. The intensity of the pinning force is determined by several factors, including the radius (*r*) and volume fraction (*f*) occupied by the secondary particles. In a three-dimensional, polycrystalline material, temperature-induced grain growth is inhibited when the Zener pinning pressure overcomes the driving force for grain growth [67]. This happens when the grain size of the main phase (*R*) reaches the Zener limit R_z_ = 4r/3f.

In order to apply the Zener pinning hypothesis to our system, we consider that the mean radius of the Ag NPs in our material is 0.75 nm [24], and that the volume fraction of Ag is *f*^−^ = ρ_TiO2_/ρ_Ag_ = (3.4 g/cm^3^)/(10.5 g/cm^3^) ≈ 0.32, thus obtaining a Zener limit R = 3 nm for the TiO_2_ NPs. Most of the Ag-TiO_2_ NPs in our film have a mean radius of 2.8 ± 1 nm (see Appendix A of the Appendix A). Since they are composed by a main TiO_2_ body partially embedding Ag NPs, as observed by previous transmission electron microscopy measurements [24], we may assume a mean radius of 2 nm for the TiO_2_ NPs, which is reasonably close to the radius predicted by the Zener limit. Moreover, since the Zener pinning is not the only force to counteract grain growth, the limit we obtained is likely an overestimation of the grain stagnation limit. The presence of pores and other defects, for example, can introduce an additional energy barrier to the grain coalescence phenomenon [68]. Additionally, the Zener pinning in 2D systems is a quite different phenomenon than its 3D counterpart. In fact, it has been suggested [69] that the Zener limit in 2D systems depends on *f*^−1/2^ rather than *f*^−1^, which means that the grain stagnation radius should decrease further as we approach lower-dimensionality systems such as thin films. Since we observe no grain coalescence for temperatures up to 500 °C, we deduce that the radius of the TiO_2_ particles in the as-deposited film is already higher than the grain stagnation limit. This result indicates SCBD as a preferred method to deposit multielement thin films with high heat resistance.

Above 600 °C, the Ag-TiO_2_ grain coarsening starts to take place, as shown in Figure 2b, where the mode radius of the GSD increases linearly from 20 nm at 500 °C to 30 nm at 900 °C. This could indicate that the driving force for grain growth has overcome the Zener pinning pressure, leading to a higher grain boundary mobility. The release of the Zener force can be explained by considering a temperature-induced modification of the Ag NPs (see also the optical behaviour described below), by analogy with a statistical mean-field model simulating the grain growth in systems with a time-decreasing pinning force [70]. In the model, the GSD evolution presents an initial stagnation of the average grain size followed by a roughly linear growth regime with time. Even though this is very close to our observations, we recall that here the pinning force decreases with every successive annealing, and not over time, hence we cannot directly apply such a model to our system. However, this approach could likely be used as a starting point for a theoretical description of many types of multielement nanogranular materials.

The RMS value follows the temperature dependence of the GSD, as shown in Figure 2c. The initial value is almost unchanged up to 500 °C, then rapidly increases by up to 25% with subsequent annealing, the sharpest increase being after the annealing step at 900 °C. The sharp increase in surface roughness is mainly due to the formation of pits and cracks, which originate during the grain coarsening of the film. The depth of the cracks ranges from 25 nm up to 50 nm, thus partially exposing the underlying silica substrate. This is compatible with the XPS data taken on this film (see Appendix A), which show the presence of SiO_2_ in the silica-deposited film annealed at 900 °C for 1 h.

### 3.2. Optical Absorption of the Films

Figure 3a,b show the optical absorption spectra as a function of the annealing temperature of the films deposited on silica and sapphire, respectively. The data are strikingly similar for the two substrates up to 600 °C. The steep increase located between 320 and 360 nm is due to TiO_2_ interband transitions. The RT spectrum broad absorption band, centered at 451.8 ± 0.8 nm on silicon and at 453.2 ± 0.8 nm on sapphire, has been observed in other Ag/TiO_2_ nanocomposite thin films [42,52] and attributed to the surface plasmon absorption of the Ag NPs, having a radius of 5–9 nm. It also worth noting that the spectrum shape is very similar to that obtained for small (1.5–3 nm) Ag/TiO_2_ core-shell NPs in solution [71].

Upon annealing the films at temperatures up to 500 °C, the maximum of the absorption band is shifted to 522.6 ± 0.8 nm and 526.4 ± 0.8 nm, for the film deposited on silica and on sapphire, respectively. If one considers the size-dependent plasmon frequency of the metal NPs, the redshift could be explained by Ag diffusion through an unmodified TiO_2_ matrix forming larger NPs. Diffusion-based coalescence between Ag NPs has been observed in Ag/TiO_2_ thin films for temperatures up to 600 °C [9]. In this temperature range, we did not observe variations of GSD (Figure 2b) nor of RMS (Figure 2c), that are measured on the film surface, which implies that the coalescence between Ag NPs doesn’t affect the Zener pinning applied to the TiO_2_ grains. This can be explained by considering the Ag diffusion along the TiO_2_ grain boundaries.

At 600 °C, the broad absorption structure almost disappears, while the absorptivity reaches 30 to 32% for wavelengths above 380 nm. This optical response modification is correlated to the beginning of the grain coarsening observed in the AFM data at 600 °C (Figure 2b,c), while TiO_2_ still appears in the amorphous phase (see the Raman spectra of Figure 4). Since the diffusion coefficient of Ag through TiO_2_ increases by an order of magnitude at 600 °C, as observed by Kulczyk-Malecka et al. [11], these observations are consistent with an Ag diffusion inside the TiO_2_ matrix. Such an event is likely modifying the Ag distribution throughout the film, and the flat absorption of the optical spectra is consistent with the presence of Ag either in atomic form, or to a very broad size distribution of Ag nanoparticles.

Further modifications of the absorption spectra are observed above 700 °C, with the films exhibiting a very different behaviour depending on their substrate. In the sapphire-deposited film (Figure 3b), the absorptivity of the film above 800 nm is steadily reduced after every annealing step, with the film becoming transparent (A = 2.4%) in the Vis-NIR range at 900 °C. As observed from the AFM data, annealing at 900 °C for 1h on this film results in the formation of Ag clusters on the surface of the film, whose dimensions (570 ± 12 nm) are too large to produce an LSPR oscillation in the visible range, justifying the absence of absorption features in this sample after annealing at 900 °C.

In the silica-deposited sample (Figure 3a), the annealing process brings to the formation of a new absorption band in the 400–600 nm range of the spectrum. The peculiar, double-peak shape of this structure can be observed in spherical isolated NPs with radius >20 nm [72], as well as in non-spherical particles (e.g., nanorods in aqueous solution) [73]. The increased Ag diffusion coefficient in this annealing temperature range would justify the observed structure by considering the formation of new Ag NPs with sizes larger than those observed at RT.

### 3.3. Substrate Effect on Silver Diffusion

The Ag amount and distribution at the end of the annealing process were investigated via SEM and EDX. Figure 4a,b show the SEM images taken after annealing at 900 °C for 1h on the films deposited on silica and sapphire, respectively, while Figure 4c,d show the corresponding Ag EDX maps. On the sapphire deposited substrate, Ag islands of 570 ± 12 nm average width are present, while no Ag signal is detected elsewhere, confirming the hypothesis of Ag diffusion through the TiO_2_ matrix that leads to the aggregation of the metal on the surface of the TiO_2_ film. The total mass of the clusters, as calculated via AFM data, is equal to 78 ± 8% of the initial mass of Ag, meaning that a significant portion of the Ag atoms have been dispersed during annealing.

On the silica-deposited film, no such islands can be observed at any step of the annealing process (see also Figure 1). Figure 4a instead clearly shows that Ag is distributed into smaller particles evenly dispersed throughout the TiO_2_ film. The overall Ag/(Ag + Ti) atomic ratio, calculated via EDX, is the same for the two substrates within the statistical error (0.20 ± 0.01 for the film on silica and 0.19 ± 0.01 for the film on sapphire). On each sample, this parameter was calculated by averaging the results obtained on four different areas, spanning a total surface of 120 × 90 μm^2^. Since EDX has a surface sensitivity of about 2 μm, this Ag/Ti ratio does not refer to the film only, but also to a portion of the underlying substrate, meaning that, despite being arranged differently, the overall quantity of Ag is roughly the same on the uppermost part of the samples.

The temperature and composition of the substrate has been shown to play a key role in determining the crystallinity and orientation of TiO_2_ [41,74] during deposition. Singh et al. [75] observed that the deposition of TiO_2_ on LaAlO_3_ (001) at 650 °C leads to the formation of anatase crystals with a (100) preferred orientation, while the deposition of the same material on Al_2_O_3_ (0001) at the same temperature leads to rutile crystals with a (200) orientation. Nevertheless, our data clearly show that the evolution of the optical and morphological properties of the films under post-deposition treatments is drastically different depending on the underlying material. Since the films were grown simultaneously by SCBD, their properties are unaffected by the substrate at deposition, depending only on the low-energy ballistic aggregation of the clusters on the surface of the deposition substrates. Moreover, all the samples underwent the same annealing steps, meaning that the difference in distribution of Ag in the two substrates is to be ascribed only to the differences in physical and chemical properties of the substrates. It is hence possible that temperature-induced crystallization of TiO_2_ grains develops in different ways depending on the thermal properties and composition of the underlying material. A substrate-dependent evolution of the morphology during annealing was previously observed in other nanogranular materials such as ZnO [76] and Pt [77] thin films, where the mean dimension and arrangement of the surface grains were different depending on substrate material. However, to our current knowledge, there was no comparative study between annealing-related transformations of TiO_2_ on different substrates. Although the accurate determination of the thermal and physical exchanges at the film-substrate interface is beyond the scope of this paper, requiring a considerably different experimental setup and sample preparation, we identified three possible factors which may play a significant role in determining the thermodynamics of the annealing and cooling process of the Ag/TiO_2_ films:

***Difference in Ag diffusivity***. Silver atoms are able to diffuse through SiO_2_ under annealing temperatures below 600 °C [78,79,80]. Above this temperature, part of the Ag contained in the film deposited on silica film would hence be able to migrate within the substrate during the annealing process, thus lowering the Ag concentration inside the film. To the best of our knowledge, we did not find any work regarding the Ag diffusion through sapphire. We, hence, suppose that this phenomenon could be far less significant in the sapphire-deposited film, thus resulting in the Ag migration towards the surface and the formation of the large islands.

***Difference in cooling rate***. The thermal conductivity of single-crystal sapphire (30.3 W/mK) [81] is 30 times that of fused silica (1.1 W/mK) [82] at room temperature. Even if the conductivity of sapphire decreases with temperature, it is always at least one order of magnitude higher than that of silica during the entire annealing process: this could play a key role during the heating and cooling cycles of the films. Viana et al. [39] have observed the formation of Ag aggregates on the surface of Ag/TiO_2_ nanocomposite films after annealing, where the dimensions of the aggregates was dependent on the cooling rates: a faster cooling rate produces bigger clusters, and *vice versa*. In our case, the higher thermal conductivity of the sapphire substrate could lead to a faster cooling process of the associated Ag/TiO_2_ film. This, according to Viana *et al*., could explain the higher cluster size of Ag in the sapphire-deposited sample. It is also worth noting that the sapphire substrate has a lower thickness compared to the silica substrate, which could further affect the cooling rate of the films. However, since the thermal conductivity of sapphire is 10 to 30 times that of silica, while the thickness differs by a factor of 2, we expect the effect of thermal conductivity to be the most important factor in determining the cooling rate of the films.

***Difference in thermal expansion coefficient***. The thermal expansion coefficient of sapphire (5.0 × 10^−6^/°C) [81] is nearly an order of magnitude higher than that of fused silica (4.1 × 10^−7^/°C) [83]. This may lead to a higher stretching of the deposited film on sapphire and hence favouring the Ag migration towards the surface of the film.

### 3.4. Effect of Silver on the Phase Transitions of TiO_2_

Nanocrystalline TiO_2_ exists in several polymorphic forms, including amorphous, anatase and rutile depending on synthesis conditions and post-synthesis heat treatments. Since amorphous TiO_2_ is thermally unstable, it transforms into anatase or rutile if heated. The temperatures of the amorphous-to-anatase and anatase-to-rutile phase transitions are different, depending on the film synthesis methods and the annealing conditions [16,23,38,84].

In order to investigate the evolution of the crystalline phase of TiO_2_, we acquired the vibrational spectra of Ag/TiO_2_ films via Raman spectroscopy after every annealing step. Figure 5a,b show Raman spectra of the Ag/TiO_2_ films on sapphire and silica for selected temperatures, respectively. At RT, the absence of the characteristic anatase and rutile Raman peaks indicate that the films are amorphous, as previously observed [85]. The weak band located at 150 cm^-1^ is commonly observed in the Raman spectra of silver oxides [86,87].

The films remain amorphous for annealing temperatures up to 600°C, while, above 700 °C, the Raman peaks of rutile at 244, 440 and 610 cm^−1^ are clearly visible in both films, as shown in Figure 5a,b. Additionally, the band at 150 cm^−1^ becomes sharper and more intense, presenting a maximum at 144 cm^−1^, which corresponds to the main vibrational mode of anatase.

One should note that for hundreds of nm thick pure TiO_2_ films grown by SCBD on silicon substrates the phase transition onsets at 400 °C [16,23], while there is coexistence of both anatase and rutile phases after 600 °C [16]. Moreover, for Ag/TiO_2_ thin films grown by pulsed cathodic arc, the phase transition temperatures depend on the Ag concentration [38]. For an Ag:Ti atomic ratio higher than 28% (equal to a mass percentage of 46%), amorphous TiO_2_ is stable for temperatures up to 600 °C, while, for T > 600 °C, there is coexistence of both anatase and rutile phases.

Therefore, the delay of the crystallization process with respects to TiO_2_ systems deposited by SCBD is likely due to the presence of Ag, given by the Zener pinning pressure applied by the Ag NPs on the grain boundaries of TiO_2_.

## 4. Conclusions

We investigated the substrate role on temperature activated processes in nanogranular Ag/TiO_2_ thin films deposited on fused silica and sapphire, thanks to the SCBD synthesis method that allowed to deposit identical films on the two different supports. The data show that annealing ends up in very different film properties depending on the substrate type. The film morphology was stable up to 600 °C, likely due to the Zener pinning pressure exerted by the Ag NPs on the TiO_2_ grain boundaries. Above this temperature, we observed grain coalescence, and the formation of both anatase and rutile phases began, accompanied by a drastic change in the film optical properties. This is explained by hypothesizing Ag diffusion through TiO_2_, thus releasing the Zener pinning. Above 600 °C, the silver distribution ended up in a completely different way depending on the deposition substrate: on silica, Ag was still present, as NPs distributed into the TiO_2_ matrix, while on sapphire, hundreds of nm wide Ag aggregates were left on top of the film. The optical properties of the annealed films reflected the different size and distribution of the Ag NPs, with the silica-deposited film presenting a broad absorption band in the visible range even after annealing at 900 °C, and the sapphire-deposited film becoming almost transparent for wavelengths above 380 nm. Three possible different causes for the observed substrate dependent behaviour were discussed.

Our findings highlight the capability of SCBD for the experimental comparison of the substrate role during post-deposition treatments, and opens up new perspectives on the thermodynamics and physical exchanges between the films and the substrate.

## Figures and Tables

**Figure 1 nanomaterials-11-02253-f001:**
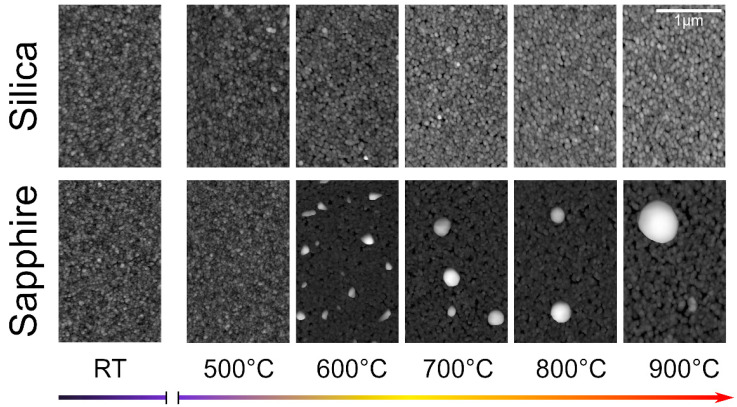
AFM images of the Ag/TiO_2_ (Ag/Ti 50–50) films on silica (**top**) and sapphire (**bottom**) as a function of the different annealing steps. Each image is 325 × 512 square pixels.

**Figure 2 nanomaterials-11-02253-f002:**
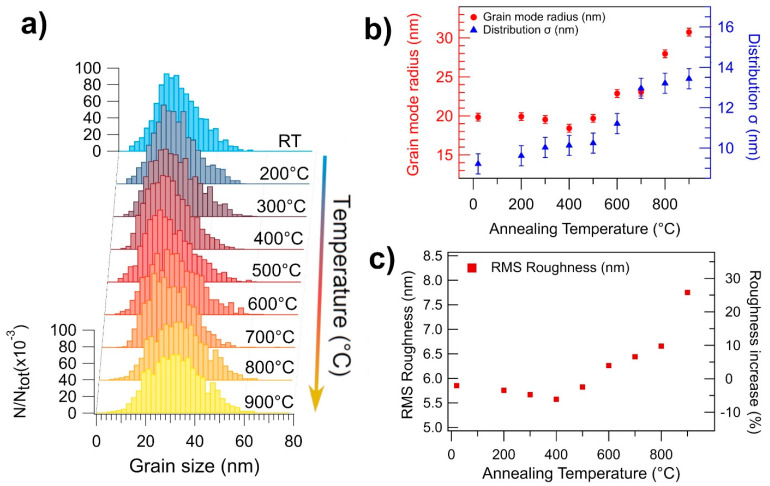
(**a**) Evolution of the GSD of the Ag/TiO_2_ (Ag/Ti 50–50) film on silica as a function of the annealing temperature. (**b**) Evolution of the mode (red dots, left axis) and standard deviation (blue dots, right axis) of the GSD as a function of the annealing temperature. (**c**) RMS roughness of the Ag/TiO_2_ (Ag/Ti 50–50) film on silica as a function of the annealing temperature. The percentage increase in RMS roughness, compared to the as-deposited film, is reported on the right axis referring to the same data.

**Figure 3 nanomaterials-11-02253-f003:**
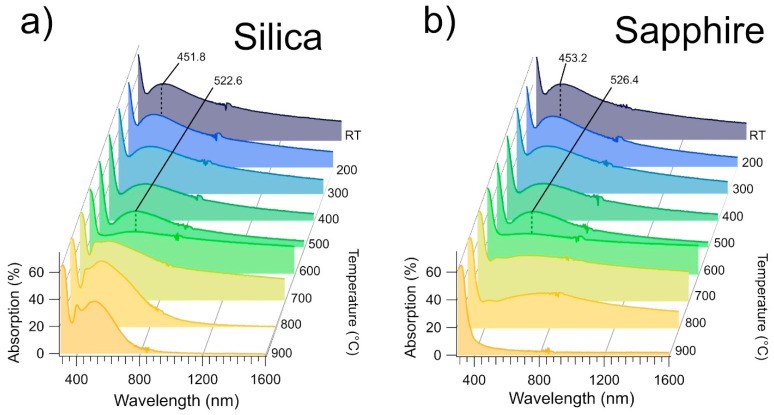
(**a**) Optical absorption spectrum of the Ag/TiO_2_ (AgTi 50–50) film on silica as a function of the annealing temperature. (**b**) Optical absorption spectrum of the Ag/TiO_2_ (AgTi 50–50) film on sapphire as a function of the annealing temperature.

**Figure 4 nanomaterials-11-02253-f004:**
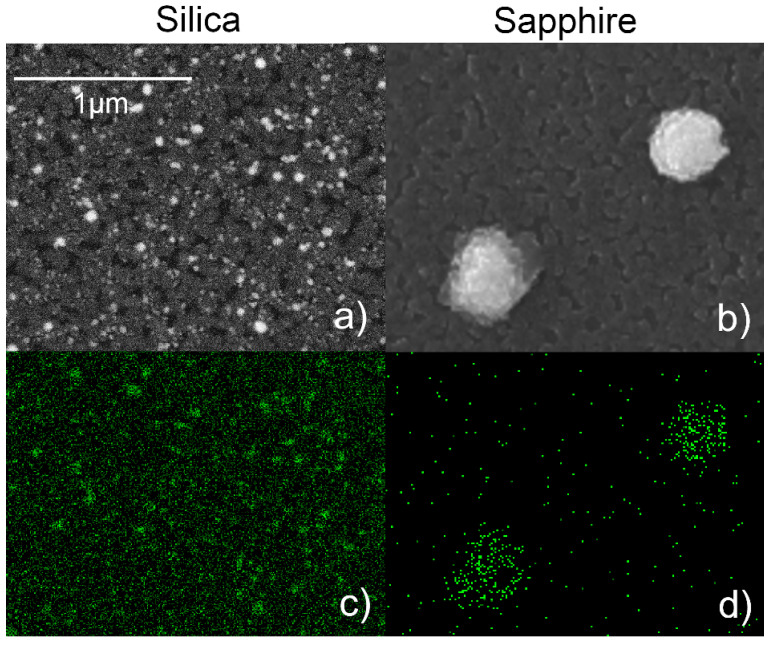
(**a**) SEM image (backscattered mode) of the Ag/TiO_2_ (AgTi 50–50) film on silica after annealing at 900 °C for 1 h. (**b**) SEM image (secondary electrons) of the Ag/TiO_2_ (AgTi 50–50) film on sapphire after annealing at 900 °C for 1 h. (**c**) EDX map of the Ag La emission acquired on the same area of (**a**). (**d**) EDX map of the Ag Lα emission acquired on the same area of (**b**).

**Figure 5 nanomaterials-11-02253-f005:**
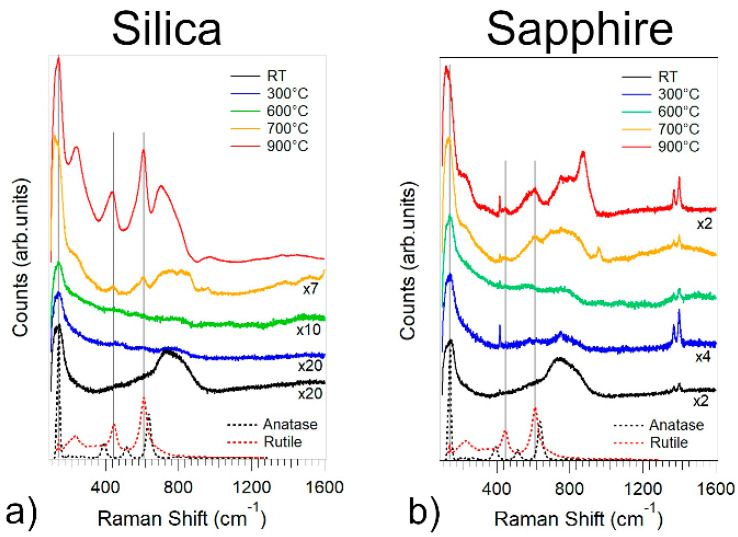
Raman spectra of the Ag/TiO_2_ film on silica (**a**) and sapphire (**b**) as a function of the annealing temperature, respectively. The dotted lines represent the Raman spectra of anatase (red) and rutile (black).

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
