# Peer review of "The Role of Substrate on Thermal Evolution of Ag/TiO2 Nanogranular Thin Films"

_nanomaterials, 2021, doi:10.3390/nano11092253_

Round 1

Reviewer 1 Report

My comments are in attached file.

Reviewer 2 Report

The manuscript reports the thermal evaluation of AgTi5050 nanogranular thin films. However, the discussion is insufficient. Several comments have to address here;

  1. In Fig. 1, the SEM images are unclear.
  2. In Fig. 4, the films are continuous films. How to identify the grain size?
  3. Cross-sectional SEM images are required.
  4. EDX spectroscopies are required.
  5. On page 3, line 6, Ag also be oxide after annealing. It not be Ag/Ti02. It would be AgTiO2.
  6. XRD patterns are required.

Therefore, I can not recommend it to publish.

Reviewer 3 Report

Manuscript ID: Nanomaterials-1314065

Title: The role of substrate on thermal evolution of Ag/TiO2 nanogranular thin films

The manuscript describes temperature dependent behavior of Ag/TiO2 nanogranular thin film deposited by gas phase supersonic cluster beam deposition (SCBD) on silica and sapphire substrates. Although not novel, the topic is interesting and worthy of investigation.

The paper could be accepted for publication after a major revision. Here are the points that the authors should consider:

  1. Authors need to emphasis the novelty and significance of the present work. Since, they have published several articles using same material and methods i.e.: doi:10.3390/nano7120442, https://doi.org/10.1021/acs.jpcc.6b00160.
  2. The information about growth time need to include in experimental section.  
  3. There are lack of information in annealing steps, author mentioned 1hr with different temperature and compared also 2hr in later part, which confuse the readers. What is the purpose of comparing 2hr?
  4. In Optical absorption spectrum, the broad absorption almost disappears for slica at 600°C, 900°C (2hr) and 900°C for sapphire. Is it due to the structural change? Why exhibits high variation? Need more explanation on this point. 
  5. Also, comparison of 2hr films may mislead the reader, authors could add more information about 2hr films or remove for better understanding.
  6. Raman and XPS could offers better bonding nature, also XPS provided for only Ag/TiO2 sample on silica. In this condition, XRD could exhibits the crystalline and structural effects, if possible authors may add XRD results.  
  7. The significance or advantage of SCBD is not clearly stated in discussion and conclusion section.

Reviewer 4 Report

Reviewer Comments

The manuscript nanomaterials-1314065 entitled " The role of substrate on thermal evolution of Ag/TiO2 nanogranular thin films" reports the influence of post-deposition annealing treatments in morphology, optical absorption, and substrate effect silver diffusion and crystal structure of Ag/TiO2 nanogranular thin films.

Please correct this part of the manuscript!

Abstract: Scientific significance of this research is not highlighted. The main work is not even well summarized either.

Introduction: The purpose of using a silica and sapphire substrate is not clear.

Material and Methods:

  • (Page 3, lines 105) “The films were deposited in medium vacuum (base pressure = 1x10-6 mbar) and then exposed to air in order to obtain complete titanium oxidation”. How did the author determine complete titanium oxidation using this method? Then how long time needs for a sample to be exposed to the air?
  • (Page 3, lines 12) Do you have etching the samples for XPS analysis?

Result and Discussion:

  • Did the author experiment use only the TiO2 sample to compare with Ag/TiO2? Using TiO2 as a sample will clearly indicate that the increasing crystallinity and phase transition of TiO2 due to high annealing temperature or Ag.
  • Could the author explain why such an island's presence just happens in the sapphire substrate, not the silica substrate?
  • In materials methods, the author explains that the annealing step was 1 hour. However, in the result discussion data, the author did anneal step for 900oC for 2 hours. Is there any consideration to add 2 hours? Furthermore, why does only silica substrate has this treatment, not for sapphire substrate? Then, could the author add this information to material and methods?
  • Why RMS roughness of Ag/TiO2 is decreased when annealed 400 oC?
  • Why is O1s binding energy shifted after annealing 900oC? How about XPS analysis for Ag and TiO2? Why is Ag peak intensity decreased after annealing 900oC?
  • The EDX results of Ag/(Ag+Ti) for the normal ratio is 0.5 (as your target described 50-50 Ag-Ti wt.%). However, the result showed only 0.2. Could the author explain what Ag/TiO2 annealing temperature is used to observe and why the ratio is decreased? How many areas did you use to observe for EDX analysis?
  • This manuscript mention that “Silver atoms are able to diffuse through SiO2 under annealing temperatures below 600°C”. However, the author used a temperature of more than 600oC also, then is that condition will influence the silver atoms? Could authors add this information to this manuscript?
  • The authors mention that Ag/silver oxides are located at 150 cm-1. However, it's hard to differentiate with the anatase phase (144 cm-1). Based on literature, the Ag peak range showed around 1000-1500 cm-1.

I suggest the following:

  • Prepare a single TiO2 sample to compare with Ag/TiO2? It is much more informative and will help for a good explanation for the author's research objectives.
  • The Raman analysis shows impurities at 750 cm-1 that gave high peak intensity, need profound observation of this peak information. Maybe this element could appear in a high ratio when the author used EDX to analyze the sample—if possible, using XRD to analyzed the crystal structure.

Round 2

Reviewer 2 Report

The referee appreciated the author’s response. However, several comments still have to address:

  1. Cross-section FESEM images are required. Alternatively, the author has any evidences to show the AgTi5050 films are nanoparticle stacks.
  2. According to Figs. 4(a) and (b), the films are continuous films, even they have a lot of holes.
  3. Grain size is not particle size. It means crystallite size.
  4. EDX elementary analysis is required. The atomic ratio is an important parameter.
  5. For the binding energy, Ag is 368.2 eV, and Ag2O is 367.9 eV. They are very close. Please recheck Fig. S4(a) again. After annealing, the peak positions are shifted.
  6. For Raman shift, how about peak (or band) at 709 (to 870) cm-1? Please discussion. The position at 150 cm-1 is corresponding to silver oxide.
  7. XRD patterns and related discussion are required.
  8. According to the author's response: "The as deposited film is made by metallic Ag grains embedded in a TiO2 matrix." Please show.
  9. The film is Ag-Ti-O film.

Reviewer 3 Report

The authors have well- addressed the reviewers concerns and the current manuscript has been improved significantly, so I recommend the paper being published as it is now.

Author Response

We are grateful to the reviewer for having read our manuscript once again and we appreciate the positive outcome of the review process.

Reviewer 4 Report

Dear Authors,

Most of the reviewer comments were included in the revised version of the manuscript. 

  1. For XPS analysis, did the author analyze on the surface or in the internal area of thin films? If detected in the inner area, how deep (nm) is the analysis? (etching time). Would you please mention in materials methods?
  2. We know that the characteristics of the TiO2 thin film were already verified in many works of literature. However, the phase transition of TiO2 will be influenced by several parameters, such as film composition (Ag added concentration), film thickness, and annealing condition mentioned in the content. It is unsuitable to refer it directly in this paper.

Round 3

Reviewer 4 Report

Accept in present form